# Participatory Approaches to Addressing Missing COVID-19 Race and Ethnicity Data

**DOI:** 10.3390/ijerph18126559

**Published:** 2021-06-18

**Authors:** Farah Kader, Clyde Lanford Smith

**Affiliations:** 1School of Public Health, University of Michigan, Ann Arbor, MI 48109, USA; 2Department of Social and Behavioral Sciences, Harvard T.H. Chan School of Public Health, Boston, MA 02115, USA; lannysmith@hsph.harvard.edu; 3Brigham and Women’s Hospital, Boston, MA 02115, USA

**Keywords:** social epidemiology, racial disparities, ethnic disparities, data disaggregation, racial identity, ethnic identity, community-based participatory research, health policy

## Abstract

Varying dimensions of social, environmental, and economic vulnerability can lead to drastically different health outcomes. The novel coronavirus (SARS-CoV-19) pandemic exposes how the intersection of these vulnerabilities with individual behavior, healthcare access, and pre-existing conditions can lead to disproportionate risks of morbidity and mortality from the virus-induced illness, COVID-19. The available data shows that those who are black, indigenous, and people of color (BIPOC) bear the brunt of this risk; however, missing data on race/ethnicity from federal, state, and local agencies impedes nuanced understanding of health disparities. In this commentary, we summarize the link between racism and COVID-19 disparities and the extent of missing data on race/ethnicity in critical COVID-19 reporting. In addition, we provide an overview of the current literature on missing demographic data in the US and hypothesize how racism contributes to nonresponse in health reporting broadly. Finally, we argue that health departments and healthcare systems must engage communities of color to co-develop race/ethnicity data collection processes as part of a comprehensive strategy for achieving health equity.

## 1. Introduction

Social, environmental, and economic oppression have well-documented links to poor health. The novel coronavirus (SARS-CoV-19) pandemic has underscored how the intersection of these hardships can lead to disproportionate COVID-19-related risk, particularly among those who are black, indigenous, and people of color (BIPOC). While having complete, accurate data on race and ethnicity among COVID-19 cases and vaccine recipients is critical for improving our understanding of health disparities, much of this information is missing in the United States [1,2].

Academics, journalists, and advocates have called upon governmental bodies across the country to report complete demographic data, as required by the Centers for Disease Control and Prevention (CDC) [2,3]. However, there are a number of barriers to data collection pertaining to the social dynamics between medical professionals, local and state health agencies, and the communities under their jurisdiction. Those racial, ethnic, and cultural groups who are most significantly impacted by COVID-19 tend to be the least likely to respond to health surveys or complete race and ethnicity questions [4,5,6]. This has significant implications for the quality and completeness of data obtained from COVID-19 vaccine registration forms and contact tracing surveys.

In this commentary, we argue that the issue of missing data will persist unless authorities invest in meaningful, participatory approaches to collecting sensitive health data. We will discuss the issues underlying nonresponse, as well as steps that health departments and community organizations may take to increase local reporting rates of race and ethnicity.

## 2. Racism and COVID-19

Since the onset of the global SARS-CoV-2 pandemic, there have been over 34,000,000 documented cases of COVID-19 in the United States, over 600,000 of which were fatal [7]. Data obtained through state contact tracing, medical and death records, and independent research demonstrate that those who are black, indigenous, and people of color (BIPOC) have experienced the highest fatality from COVID-19 nationwide [8,9,10]. Basset et al. find that the total black population has 3.6 times the risk of dying when compared to the white population after age adjustment. Among 35–44-year-old black Americans, there is a nine-fold risk of COVID-19 fatality compared to whites in that age group [11]. In California, other ethnic group-specific health disparities are more evident due to relatively large Hispanic/Latino, Asian, Native Hawaiian, and Pacific Islander populations [12]. For example, Asian Americans accounted for 52% of COVID-19-related deaths in San Francisco despite constituting only 13% of the cases [13]. While exact differences in risk differ across methodologies and study samples, the disproportionate impact of COVID-19 on BIPOC remains consistent [14,15,16,17].

### 2.1. Inherited Access to Healthcare

Health risks posed to BIPOC populations during the pandemic correlate with structural, interpersonal, and other forms of racism that account for health and economic disparities broadly [18,19]. Critical race theory postulates that these inequities result from structural racism embedded in laws and institutions over time. For example, centuries of legalized slavery, segregation, and race-based discrimination against black Americans contribute to a stark racial wealth gap [20]. Vast differences in inherited wealth and opportunities for economic mobility between racial groups over time also translate to a multi-dimensional health gap [21,22]. Pathways from racial discrimination to poor physical health account for COVID-19-specific mortality disparities.

One contributor to this relationship is the restricted access to affordable, quality health care for BIPOC populations, which can be due to cost, limited cultural and linguistically appropriate services, and discrimination from health care providers [15,23,24,25]. In addition, a lack of health insurance, a necessary component of access to the US health care system, disproportionately affects black and Hispanic/Latino Americans [26]. The inability to afford adequate health insurance coverage and limited transportation options to medical facilities are significant barriers for low-income COVID-19 patients seeking treatment.

### 2.2. Health Inequities Related to COVID-19 Morbidity and Mortality

There are links between different forms of racism and long-term stress, depression, and other covariates of chronic illnesses such as diabetes and hypertension [27,28]. Structural racism, including housing and employment discrimination, can lead to poorer housing conditions and subsequently high exposure to industrial pollution. These conditions cause disproportionately high rates of asthma and chronic obstructive pulmonary disorder [29,30,31]. As such, BIPOC groups have more frequent chronic comorbidities that exacerbate the risk of COVID-19 morbidity and mortality [32].

### 2.3. COVID-19 Exposure Disparities

Broad racial disparities can additionally determine whether or not one is exposed to SARS-CoV-19 in the first place. Hispanic/Latino, black, and Asian Americans are all more likely to live in multigenerational households than whites, due to both financial circumstances and cultural values, increasing viral exposure and transmission [33,34]. Furthermore, the conditions of prisons and immigrant detention centers must be considered from a racial justice standpoint; the immigrant detention and incarceration in the US disproportionality impacts black Americans and other people of color [35]. These institutions amplify the risk of COVID-19 morbidity and mortality through extreme sanitation issues, crowding, food insecurity, and a range of other causes [36].

Occupational risks also disproportionately affect those who are BIPOC. Black and Hispanic/Latino Americans in the United States are overrepresented in service sector jobs that are generally considered essential during the COVID-19 pandemic and provide fewer work-from-home options [37,38]. Workers in these positions are unable to follow public health guidelines, such as social distancing, risking the likelihood of exposure to the virus [14,39].

### 2.4. Vaccine Allocation Inequity

Those facing systemic oppression stand to benefit from the protection immunization offers but many social factors inhibit access to COVID-19 vaccines. Throughout the country, registration for a COVID-19 vaccine dose requires computer skills and access to technology. This isolates those who have low income and/or are English-language limited. A restricted number of vaccine doses per county also limits the number of possible vaccination sites in a given region. Those who do not have a personal vehicle or other means of transportation may not be able to reach a site where appointments are available. Such challenges may act as external barriers to vaccination for those who may otherwise be willing to receive the COVID-19 vaccine [40,41,42].

## 3. Missing Race/Ethnicity Data

Limited national data on COVID-19 morbidity and mortality by race/ethnicity is a public health concern. Presently, more than half of the CDC’s reported case data are missing racial and ethnic identifiers, with state health departments and healthcare systems reporting race data that greatly vary in completeness [2].

Similarly, there are significant gaps in COVID-19 vaccine administration data when stratified by race and ethnicity. Vaccine data completeness is necessary for determining equity in vaccine access and allocation. The first among those vaccines to protect against COVID-19 was approved for emergency use in December 2020. By February 2021, federal researchers found that nearly half of coronavirus vaccine recipients had missing race and ethnicity information [1]. However, analyses of the available race/ethnicity information “show that the shares of vaccinations among Black and Hispanic people are lower compared to their shares of the total population” [1,43]. A more robust and inclusive understanding of this disparity requires addressing the significant race/ethnicity data gap.

## 4. Contributors to Nonresponse

### 4.1. Definitions of Race and Ethnicity

While discrimination can shape the health outcomes of a wide range of distinct and overlapping racial, ethnic, religious, and cultural communities, most health research and reporting capture disparities between the federally recognized racial and ethnic groups. The Office of Management and Budget (OMB) determines the national standard on race and ethnicity. These categories are required on a variety of forms, including the US census, employment applications, medical forms, clinical trials, and school tests. These were last revised in 1997, when the OMB deliberated existing racial categories, using input from special interest groups and other agencies [44]. While they are required for use by federal agencies, they are not binding on states [45].

Contemporary definitions of race and ethnicity may not include all identities, while many individuals may identify with more than one racial or ethnic group. However, the OMB standard fails to account for all of these dimensions of race and ethnicity. For example, the standard defines the white race to include individuals from “the Middle East and North Africa.” The Arab American Institute has advocated for a special ethnic category due to the multiracial nature of Southwest Asia and North Africa and a growing US population descending from the region [46,47]. In San Diego County, California, the zip codes with the highest rates of COVID-19 cases, hospitalizations, and deaths overlap with the area’s Arab American enclaves [48,49]. Thus, reliance on OMB categories can mask disparities that require attention from local health departments.

This context partly explains why there is a high rate of nonresponse on standardized race/ethnicity survey questions, including COVID-19 case data [6,50]. The language used to describe ethnicity and race is not universal, particularly among those with intersectional backgrounds. Furthermore, members of certain vulnerable populations may be opposed to identifying with a racial group with whom they may not share linguistic, ethnocultural, and/or physical characteristics [51]. On the 2010 census, such a large portion of the Hispanic population identified as “Some Other Race” that this became the third-most selected category in the US, demonstrating people’s resistance to labeling themselves on official documents using OMB standard language [52].

### 4.2. Racial Misclassification

The aforementioned discrepancies between government-defined racial categories and individuals’ perceptions of race and ethnicity can lead to racial misclassification. For instance, multiracial individuals may not indicate all aspects of their identities on surveys due to limitations in the race/ethnicity options presented. This kind of disagreement regarding racial/ethnic data may result in leaders’ misunderstanding their local demographics, even when nonresponse rates are low [53,54].

Furthermore, racial misclassification by healthcare personnel and other individuals administering in-person health surveys is a concern. Groups who experience high rates of racial misclassification by those who perceive them to be of a “different” racial/ethnic group include, but are not limited to, Afro-Latinos, Native Americans, and Arabs. There is evidence that this type of misclassification results in psychological distress for misidentified persons in addition to inaccurate reporting of diseases such as cancers and adverse birth outcomes [55,56,57,58,59].

### 4.3. Mistrust of Government

Because of mistrust between BIPOC populations and public health authorities, some may not be comfortable disclosing racial and ethnic identity information to contact tracers, healthcare systems, and other collectors of COVID-19 case information. Much of this mistrust stems from known ethics violations in US-based medical research. Examples of medical and scientific professionals’ exploitation of communities of color include the Tuskegee syphilis studies on African American men, early birth control trials on poor Puerto Rican women, and a sham vaccine distribution program in Pakistan used by the US Central Intelligence Agency for counterterrorism endeavors [60,61,62]. These issues have been attributed to people of color, particularly black Americans, demonstrating lower rates of acceptance and trust in novel COVID-19 vaccines when compared to other groups [63].

The history of exploitation in marginalized groups can be a barrier to not only vaccine hesitancy but also race/ethnicity data accuracy and completion. Responding to government inquiries on ethnic group or country of origin is a sensitive issue for members for frequent targets of government surveillance, forced disappearances, and deportations. This includes religious groups that largely overlap with BIPOC populations, like Muslims and Sikhs [64]. Places of worship and community centers that primarily serve African Americans and individuals of North African, Southeast Asian, and Southwest Asian descent are frequently surveilled and profiled as potential terrorist organizations by the US government [65,66]. Similarly, immigrants and asylum seekers from Mexico and Central America are disproportionately targeted in raids and arrests by US Immigration and Customs Enforcement [67]. These experiences may make BIPOC groups more hesitant to respond to COVID-19 surveillance efforts, which entail querying cases about their whereabouts and activities for contact tracing purposes.

## 5. Recommendations

### 5.1. Data Disaggregation

As discussed, the OMB standards defining race and ethnicity do not recognize the overlap between racial and ethnic groups, nor do they account for the variation that exists within them [68,69]. In response to this issue, health equity advocates and researchers have called for racial/ethnic group disaggregation as an important method for epidemiological surveillance [70,71]. Systemic examination of COVID-19 risk at a level that is more granular than broad race categories can elucidate drivers of health disparity that are connected to ethnicity and culture.

Because direct actions to address health disparities are most likely to occur at a local level, local and state health departments should consider data on the subgroups within their jurisdictions. The 1997 OMB standard includes a recommendation for local governments to create additional racial/ethnic categories for populous ethnic groups within in their jurisdictions, “provided they can be aggregated to the standard categories” [72]. On a national scale, certain ethnic populations may not be large enough to calculate reliable case rates, mortality rates, and other health statistics. However, the local jurisdiction may benefit from counting distinct ethnocultural communities whose health behaviors, chronic conditions, and other characteristics may heighten COVID-19-related risk.

Institutions that advocate for BIPOC communities have supported community-led initiatives to disaggregate racial categories. A report on data disaggregation, commissioned by the Robert Wood Johnson Foundation and published by PolicyLink in 2018, involves a collaboration between the Urban Indian Health Institute, the Asian & Pacific Islander American Health Forum, and the Black Women’s Health Imperative, and others. The report outlines several methods for collecting and analyzing data about race and ethnicity and government policies that enable data disaggregation [70]. The document also summarizes existing methods for avoiding racial misclassification and measuring multiracial populations. Similarly, the Institute of Medicine’s Unequal Treatment: Confronting Racial and Ethnic Disparities in Health Care, asserts that “efforts to address disparities in care must acknowledge the significant heterogeneity within each of the federally defined racial and ethnic groups” [73].

In January 2021, *Population Research and Policy Review* published a special issue entitled, “The Critical Role of Racial/Ethnic Data Disaggregation for Health Equality.” This issue dedicates one article for each of the five OMB standard racial/ethnic categories, highlighting their limitations in the context of health research and the COVID-19 pandemic. The literature reviews in this issue underscore the national standard’s failings in evaluating the effect roles of racism and social oppression on health. Issues such as the homogenization of the Asian, Native American, and black populations, as well as the inclusion of multiracial Middle Eastern and North African groups in the white category, are among the many issues that mask racism’s multifaceted impact [68].

### 5.2. Partner Engagement

Health authorities must engage community organizations when deliberating methods of data disaggregation and other strategies for improving data quality and completeness [70,74]. Decision-making processes that engage those who are most impacted by COVID-19 will not only clarify appropriate strategies for collecting sensitive information but also create community acceptance and willingness to participate in needed health surveillance [70]. Community advisory members can also help technical experts to expand conceptualizations of race/ethnicity and survey populations that are truly representative of the community as a whole.

To help bridge health officials to the communities they serve, CDC Vaccine Program’s Interim Playbook for Local Jurisdictions recommends using partner agencies and organizations to get accurate estimates of critical populations and their needs. Partners may be grassroots organizations, federally qualified health centers, faith-based groups, and cultural organizations that are equipped to inquire about racial and ethnic identity in socially oppressed communities [75]. Using the list of potential partner classifications from the Playbook to establish points of contact, local health departments can leverage partnerships with trusted community organizations to aid tailored communications and inform appropriate data collection strategies.

### 5.3. Examples in Practice

The annual American Community Survey (ACS) provides another model for identifying more granular ethnicities. While the question is multiple-choice, there are a large number of coded responses that would account for the majority of self-identified ethnicities in the US population. Incorporating these comprehensive answer options into state contact tracing systems would be inclusive, although potentially a burdensome level of detail for disease surveillance. It is also difficult to include these data in predictive models and other analyses due to small sample sizes [70,76].

Given the analytical challenges of open-ended self-identification for large-scale data collection, it is most feasible for health departments to disaggregate standardized racial categories into the most locally prevalent subgroups [69,71]. Some local jurisdictions have already implemented this practice when tracking COVID-19 cases. The Hawaii State Department of Health is the first state to publicly present disaggregated data on the state’s most populous Asian subgroups [77]. According to their state dashboard, Filipinos and other Pacific Islanders are much more likely to have a confirmed case of COVID-19 when compared to Native Hawaiians and other Asian ethnic groups with whom they are often grouped [78]. This data supports numerous studies that find that great health and economic diversity among Asian and Pacific Islander subgroups; therefore, they must be studied as separate ethnicities [68,79].

Many frameworks of community participation in official health decision-making exist, and some have been applied to reduce COVID-19 disparities. For example, some city and county health departments across the US established advisory groups, community focus groups, and public deliberation mechanisms to inform health initiatives and procedures [80]. Applying these strategies to data-related decision-making may address missing race/ethnicity information at the local level and help identify the most appropriate and viable data collection mechanisms.

## 6. Challenges

Disaggregation is beneficial because it resists the homogenization of disparate groups, urging health authorities to view racial inequity as a product of racism rather than a function of race, as defined by the OMB [51]. However, efforts to disaggregate broad racial categories should not serve to minimize the significant effect of race, particularly blackness, which exists across many ethnicities [81,82]. Maintaining the ability to collapse members of more granular ethnic identities into a larger race is important for continuing to measure the effects of anti-black racism on health.

Health departments must be mindful of the potential for disaggregated data on COVID-19 case and vaccination rates to stigmatize specific groups. This is a particular concern for Asian Americans and Pacific Islanders, who have faced a rise in racially motivated violence during the COVID-19 pandemic due to reports of the virus’s origin in Wuhan, China. Changes in race and ethnicity data categories may necessitate training interpreters and collectors of data [83]. Health departments also have a responsibility to communicate racial/ethnic data in ways that do not perpetuate stereotypes or dehumanize groups that have COVID-19. Organizations such as the World Health Organization have established best practices for disseminating ethical health communications [84].

A potential limitation of disaggregating populous subgroups is the “enclave effect,” which refers to the health outcomes of regionally concentrated ethnic groups observed in studies. Some ethnic enclaves can have unique characteristics that result in positive health outcomes and reduced disparities, such as a concentration of well-established, culturally and linguistically competent health and social services [85]. As such, the findings of these studies cannot be broadly applied. Conversely, more granular inquiry of race/ethnicity can result in disaggregating groups into small sample sizes, which may not be useful for reporting and analysis purposes. Analytical methods, including field research and pooled datasets, should be explored in order to address small populations that are underrepresented in health disparities data [70].

While community input is vital for health, within-group discrimination and domination patterns including sexism, classism, homophobia, colorism, and other dynamics may lead to a misrepresentation of community needs if such correlates of health are not deliberated [86,87]. Health departments must seek numerous perspectives from priority populations and refrain from overreliance on the views of select organizational leaders and others in positions of power. Focus groups that recruit from various settings may inform more nuanced data collection strategies and analytical methods.

Finally, political will is required to invest in collecting more detailed race and ethnicity data among constituents [79]. Additionally, community partner engagement with local health departments requires a significant investment of time and resources, which may be scarce in situations of urgent health and economic emergency. Certain data disaggregation methods, such as textual analyses of open-ended responses, may be required. Relationships with universities and large non-governmental institutions may be necessary to provide leadership, coordination, and administrative support in facilitating an open dialogue on racial and ethnic data collection. It is also possible that such partnerships will not produce immediate results, rather they will provide foundations for gradual improvements in community-government relations that result in higher-quality health data in the long-term.

## 7. Conclusions

Missing cultural, ethnic and racial data serves to disadvantage BIPOC groups by obscuring the extent of COVID-19 vaccination, prevention, and treatment inequality. Yet, calls for the CDC and local health departments to address this problem often neglect to consider the underlying reasons for missing data, especially among those who are BIPOC. Subsequently, we propose that health authorities apply existing models of community participation in decision-making to data collection procedures in order to improve the quality and completeness of racial/ethnic data. In the short-term, shifting the manner in which health surveillance instruments inquire about this information may allow local health departments to better understand the hard-to-reach populations within their jurisdictions. In the long-term, policies that require partnerships between health departments, providers, and community and faith-based organizations can develop more appropriate, equitable strategies for measuring health disparities. While changes to data collection and analyses are necessary for understanding the health of diverse populations, health leaders must take steps to ensure these processes service communities of color without enabling stigmatization or dismissing the heterogeneity within identified racial and ethnic groups.

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
