# Peer review of "Participatory Approaches to Addressing Missing COVID-19 Race and Ethnicity Data"

_ijerph, 2021, doi:10.3390/ijerph18126559_

Round 1
Reviewer 1 Report
This commentary is timely and well-written. I have no major comments. One minor suggestion is that the authors could discuss about the cost related to additional racial/ethnic categories (although it could be difficult to quantify) and open-end responses (which may result in additional cost in processing the text inputs).
Author Response
Thank you to the reviewer for their evaluation of our draft. In response to their suggestion regarding the limitations of disaggregation and open-ended response, we have elaborated on potential added costs and resources for more thorough data collection and analysis.
Reviewer 2 Report
The manuscript is not a research article but a very thoughtful commentary on the importance of including race and ethnicity data in Covid-19 surveillance tools and the reasons why it might be difficult to do so considering underlying racial inequities in the U.S. particularly when it comes to accessing healthcare.
The argument is overall well-organized and covers the issue extensively (thus giving it great potential as a future reference for more specific research).
The following suggestions could further improve the comprehensiveness of the paper and help nuance the discussion section:
We recommend reorganizing the “Racial disparities and Covid-19” section by showing how three separate dynamics in racial disparities (1) inherited access to healthcare 2) current poorer health status + 3) higher risk of exposure to Covid create negative synergies explaining why minorities are disproportionately affected by Covid morbidity and mortality.
Right now, the paragraph reads a bit like a list of determinants of health disparities, but the interrelations are not clearly outlined.
In addition, we believe there would be merit in nuancing the use of adjective “racist” in this section. While we do not question the racist origin of these determinants, it might be important to clarify how, even in the absence of intentionally racists policies currently, racial inequalities are compounded and replay dramatically in the Covid pandemic, and therefore require additional and specific work to be overcome (i.e. simply having “non racist” policies and practices is not enough).
It might be useful to expand further on the necessity of race and ethnicity data as crucial to understand not only who is under-vaccinated but also why, for example by clarifying health inequities among certain minorities or specific determinants of vaccine hesitancy.
From a pragmatic perspective, it might be interesting to indicate that “race / ethnicity” data are often “assigned” by the data collector in charge at the immunization site (who can be a nurse, hospital or city staff, or volunteer registering patients) and that this assignment is often decided based on visual cues rather that self-identification. (This is for example problematic in areas with high Black Hispanic population who are miscategorized as “African Americans”).
One limitation that should be mentioned however is the risk of singling out certain communities and thus putting them at risk of stigma. This was especially relevant for testing data but may also replay in the case of immunization.
Also, it would seem important to specifically discuss the anti-Asian racism associated with the Covid-19 pandemic and how improved statistic quality may be relevant to addressing some of the health inequalities currently faced by these populations.
Finally, while community input might be a relevant strategy to address poor data quality (or rather specificity when it comes to race and ethnic data), the authors might also want to refer to the literature suggesting that discrimination and domination patterns existing within racial and ethnic minorities as well that might not be obvious to individuals analyzing vaccination data.
Overall, it might be useful to argue for the relevant level of specificity of race / ethnicity statistics with the goal of more appropriately serving minorities without either diluting the data or risking stigma for specific groups.
Author Response
Thank you to the reviewer/s for their thoughtful and thorough comments.
We have incorporated their suggestion to reorganize the section on Racism & COVID-19 and included subheaders to aid readers’ understanding of the arguments outlined and the reasons for them. We also agree that the adjective ‘racist’ may be viewed as vague as written and have provided definitions of critical race theory to provide context.
Because the purpose of the paper is to provide an alternative perspective on data nonresponse/inaccuracy and provide practical solutions to that problem in this urgent health crisis, we have attempted to keep this section brief. Our purpose for Section 2 is to affirm the evidence for racial health disparities rather than give a detailed discussion of the reasons for those disparities. There are many complex factors that contribute to the health inequities listed in this first section, and we hope that our citations can be helpful references for those who want to explore those underlying issues further.
While we did not explicitly use the term ‘vaccine hesitancy’ in our paper, we have a section on government mistrust and have added a sentence to further explicate that these issues disproportionately affect willingness to be vaccinated among people of color.
We would also like to thank the reviewers for pointing out the fact that race/ethnicity misclassification in medical settings is a problem evident in the literature. We have added a section on this issue and provided the appropriate citations.
In our discussion of the limitations of disaggregated data, we have included the potential for stigmatization of marginalized groups for the spread of disease with specific reference to anti-Asian racism. We have also elaborated on limitations of community input when dominant voices within ethnic/racial groups overshadow others, an important factor the reviewers have raised. Our conclusion further emphasizes the idea that disaggregation should be used to service communities of color without ignoring the heterogeneity within them or stigmatizing them.